## Research Article

discrimination; mental health; mental illness; stigmatization; self esteem; co-production; lived experience; experts by experience

**Corresponding author:**
Petra C. Gronholm;
Email: petra.gronholm@kcl.ac.uk

# Establishing partnerships with people with lived experience of mental illness for stigma reduction in low- and middle-income settings

Gurucharan Bhaskar Mendon[1] , Dristy Gurung[2,3], Santosh Loganathan[4], Sisay Abayneh[5,6], Wufang Zhang[7], Brandon A. Kohrt[8], Charlotte Hanlon[2,6] , Heidi Lempp[9] , Graham Thornicroft[10] and Petra C. Gronholm[10]

[1]Department of Psychiatric Social Work, National Institute of Mental Health and Neuro Sciences, Bangalore, Karnataka, India; [2]Health Service and Population Research Department, Centre for Global Mental Health, Institute of Psychiatry Psychology and Neuroscience, King's College London, London, UK; [3]Transcultural Psychosocial Organization Nepal, Kathmandu, Bagmati, Nepal; [4]Department of Psychiatry, National Institute of Mental Health and Neuro Sciences, Bangalore, Karnataka, India; [5]College of Education and Behavioural Studies, Bale Robe, Madda Walabu University, Robe, Ethiopia; [6]Department of Psychiatry, School of Medicine, College of Health Sciences, Addis Ababa University, Addis Ababa, Ethiopia; [7]Institute of Mental Health, Peking University Sixth Hospital, Beijing, China; [8]Department of Psychiatry, Center for Global Mental Health Equity, The George Washington University, Washington, DC, USA; [9]Centre for Rheumatic Diseases, Department of Inflammation Biology, School of Immunology and Microbial Sciences, Faculty of Life Sciences and Medicine, King's College London, London, UK and [10]Health Service and Population Research Department, Centre for Global Mental Health and Centre for Implementation Science, Institute of Psychiatry Psychology and Neuroscience, King's College London, London, UK

## Abstract

Social contact refers to the facilitation of connection and interactions between people with and without mental health conditions. It can be achieved, for example, through people sharing their lived experience of mental health conditions, which is an effective strategy for stigma reduction. Meaningful involvement of people with lived experience (PWLE) in leading and co-leading anti-stigma interventions can/may promote autonomy and resilience. Our paper aimed to explore how PWLE have been involved in research and anti-stigma interventions to improve effective means of involving PWLE in stigma reduction activities in LMICs. A qualitative collective case study design was adopted. Case studies from four LMICs (China, Ethiopia, India and Nepal) are summarized, briefly reflecting on the background of the work, alongside anticipated and experienced challenges, strategies to overcome these, and recommendations for future work. We found that the involvement of PWLEs in stigma reduction is commonly a new concept in LMIC. Experienced and anticipated challenges were similar, such as identifying suitable persons to engage in the work and sustaining their involvement. Such an approach can be difficult because PWLE might be apprehensive about the negative consequences of disclosure. In many case studies, we found that long-standing professional connectedness, continued encouragement, information sharing, debriefing and support helped the participants' involvement. We recommend that confidentiality of the individual, cultural norms and family concerns be prioritized and respected during the implementation. Taking into account socio-cultural contextual factors, it is possible to directly involve PWLEs in social contact-based anti-stigma interventions.

## Impact statement

Establishing partnerships with people with lived experience (PWLE) of mental illness is essential for reducing stigma in mental health across the world. Involving PWLEs in stigma reduction efforts through strategies based on social contact is an effective and recommended approach to reduce mental health-related stigma. Additionally, involving PWLEs in such activities can benefit in other ways, such as improved self-confidence and self-esteem. However, there has been little written on how to practically implement this in economically underserved settings and in different cultural contexts. Across low- and middle-income countries (LMICs), limited access to mental health care, varying protections of human rights, limited opportunities for empowerment of PWLE and their caregivers and different explanatory models of mental health conditions provide contextual challenges requiring different approaches. To begin addressing this gap, this paper reflects on researchers' and mental health experts' perspectives on establishing partnerships with PWLEs across four LMIC countries as part of efforts to reduce mental health-related stigma. This is the first study of its kind to present a number of examples of anti-stigma research involving partnerships with PWLEs in LMICs. Hence, this paper provides important insights on the practical challenges in implementation and reflections to guide future anti-stigma activities and strategies to ensure PWLEs are meaningfully involved with anti-stigma interventions and research.

## Introduction

Social contact refers to the ways in which people with lived experience (PWLE) of mental health conditions have direct or indirect contact with people who do not have such experiences. Direct (e.g., face-to-face, individual-to-individual) and indirect (e.g., active or passive interaction, e-contact or imagined contact, such as recorded personal testimonials and narratives of lived experience and recovery) social contacts can effectively reduce stigma and discrimination (Thornicroft et al., 2016; White et al., 2021; Thornicroft et al., 2022) and can reduce self-stigma through empowering individuals with lived experience (Corrigan et al., 2013). In addition to involving PWLEs in anti-stigma activities through social contact strategies, PWLEs' involvement can be extended to other aspects of mental health care and anti-stigma activities. Indeed, PWLE is crucial in reducing stigma and promoting mental health worldwide (Thornicroft et al., 2022; Sartor, 2023). The involvement of PWLE can also bring unique expertise in developing mental health policy and legislation, planning and designing mental health programs, and service provision (Lempp et al., 2018).

Involving PWLE as partners must be equitable, including contributing to decision-making, transparency, and appropriate compensation at regional, national and international levels (Sartor, 2023). However, evidence from low- and middle-income countries (LMICs) shows that there is often little or no involvement of PWLE in mental health-related research or anti-stigma activities. For example, lack of mental health awareness, stigma associated with mental illness, minimal empowerment of PWLE, structural barriers, or lack of clarity on involving PWLE in stigma reduction or policy development (Samudre et al., 2016; Stuart, 2016; Petersen et al., 2017; Lempp et al., 2018). This contributes to the gap in understanding how PWLE can be involved in anti-stigma activities, considering that most of the research evidence is from high-resource western countries, which might not be transferable (Ryan et al., 2019; Kohrt et al., 2021). However, there is emerging evidence also from LMIC settings that stigma-reduction approaches focused on social contact through PWLE involvement can be effectively delivered (Thornicroft et al., 2022). This has been demonstrated, for example, through Time to Change Global, where social contact-based stigma reduction principles from England's Time To Change anti-stigma program were successfully delivered in Ghana and Kenya (Potts and Henderson, 2021). Also studies like Reducing Stigma among Healthcare Providers (RESHAPE) in Nepal, and the Systematic Medical Appraisal, Referral and Treatment (SMART) Mental Health project in India have shown that stigma reduction through social contact and PWLE involvement was feasible despite some challenges (Maulik et al., 2019; Kaur et al., 2021; Kaiser et al., 2022; Gurung et al., 2023; Rai et al., 2023). Notably, the SMART Mental Health project reported positive effects of their anti-stigma campaign even at 2-year follow up (Maulik et al., 2019). To expand on this evidence-base, the Lancet Commission on Ending Stigma and Discrimination in Mental Health emphasized that the involvement of PWLEs should be an integral feature of stigma reduction and promotion of mental health globally through culturally appropriate social contact strategies (Thornicroft et al., 2022). Furthermore, PWLEs are key stakeholders who need to be supported to lead or co-lead interventions using social contact (Gronholm et al., 2023, 2024; Sartor, 2023; Thornicroft et al., 2022).

The aim of this paper is to explore the factors that contribute to the active involvement of PWLE in stigma reduction in LMICs (Hanlon, 2017; Lempp et al., 2018; Kaiser et al., 2020).

## Methods

A qualitative collective case study design (Baxter and Jack, 2015) was adopted to address the knowledge gap on engaging PWLE in stigma reduction. The case studies were selected from four LMICs. Specifically, these studies were case study 1: Experience using Photovoice in Nepal: a voice to be seen (Nepal); case study 2: A social contact-based education intervention to reduce stigma among Community Health Care Staff in Beijing, China; case study 3. Experience from "Stories Against Stigma" – a Walking Tour of a Tertiary Mental Health Care facility, Bengaluru, India; and case study 4. Using participatory action research for service user and caregiver involvement in mental health system strengthening in primary care in Ethiopia. Further details on these studies are provided in Table 1. Ethical approval was granted for this work at the respective institutions.

For the reflections reported on in this paper, a team in each site shared their thoughts on i) a detailed local situation analysis, ii) anticipated challenges in involving PWLE, iii) experienced challenges, iv) strategies to overcome these challenges and v) what they have learned from PWLE and what they would recommend.

These case study examples are provided below, followed by reflections on key insights and commonalities to identify recommendations for mental health stigma reduction efforts in the future that are meaningful partnerships with PWLEs.

## Results

### Case Study 1. Experience using photovoice in Nepal: a voice to be seen

#### Background to the work

In Nepal, stigma poses a significant barrier to mental health treatment access (Luitel et al., 2017). The Transcultural Psychosocial Organization (TPO) collaborated with PWLE in projects like RESHAPE, a stigma reduction initiative. RESHAPE involves PWLE sharing recovery stories through Photovoice, a participatory research approach using photography (Kohrt et al., 2020; Rai et al., 2023). Trained PWLE then go on to co-facilitate mental health training for primary healthcare workers, where they share experiences and dispel myths. PWLE receives compensation, including 1,000 Rupees per session, travel reimbursement, and food. This approach effectively addresses stigma among healthcare workers.

Ethical clearance for this work was obtained from the Nepal Health Research Council (Ref: 441/2020).

#### Anticipated challenges

In intervention, we had anticipated that disclosure of mental illness would be a major challenge for most of the PWLE as part of the recovery narrative. In the community, mental health conditions were much stigmatized and identifying oneself living with this condition could lead to job loss, exclusion from family and community, and problems in marriage. Secondly, as the photovoice training was of 10–12 sessions and 5–6 h each, we anticipated a challenge in participation of PWLE as it could disrupt their household or fieldwork, jobs, or other chores.

#### Experienced challenges

As anticipated, some PWLEs were discontinued or dropped out. An unforeseen challenge was that due to fear of increased stigma from disclosure, PWLE had to make excuses to their family

**Table 1.** Overview of anti-stigma programs

| Case study | Type of stigma targeted | Details | Impact |
|---|---|---|---|
| 1 Experience using Photovoice in Nepal: a voice to be seen | Public Stigma, Family stigma | Stigma attitudes of primary healthcare workers were targeted through social contact with photovoice–trained PWLE. For some PWLEs, family stigma was also addressed on an *ad hoc* basis. | Through a pilot cluster–randomized control trial, improved attitudes of primary healthcare workers and improved detection of cases at primary healthcare facilities were seen in the intervention arm compared to implementation as usual arm. On occasions, at the PWLEs request, counselors facilitating the program provided psychoeducation or family counseling to PWLEs' family members who were stigmatizing toward them. However, this was not formally evaluated as a part of the training. |
| 2 A social contact–based education intervention to reduce stigma among community health care staff in Beijing, China | Public stigma | Public attitudes toward primary care and community health care staff; attitudes and intended behavior to people with mental health conditions | RCT trial, evaluated by scales. The "education and contact" group showed a significantly greater improvement in clinicians' attitudes and intended stigma–related behaviors than the "education only" group, but the between group differences disappeared at 1 month and 3 months follow–up points. The positive effects on stigma levels (knowledge, attitudes and behaviors) in both groups were sustained at 3 months. |
| 3 Experience from "Stories Against Stigma" – a Walking Tour of a Tertiary Mental Health Care facility, Bengaluru, India | Public stigma | Public attitudes toward a tertiary mental health care center; stigma associated with the medical model of treatment for mental health conditions | In the pre–assessment, 36.4% took part in the health care facility tour to know more about mental illness, about 27.3% were interested in knowing about the mental health care facility, and the remaining 27.3% wanted to improve their knowledge. In post assessment, the majority (79.4%) agreed that the tour was responsible for bringing about a positive shift in their attitude toward mental health. |
| 4 Using participatory action research for service user and caregiver involvement in mental health system strengthening in primary care in Ethiopia | Self–stigma, public stigma | Public stigma among health professionals, self–stigma among service users and caregivers, and public stigma through community perceptions about service user and caregiver involvement in mental health system strengthening. | In our qualitative studies, the service users and caregivers reported a sense of achievement being part of the research team, contribution and agency for establishing service user association, improved opportunity to get out of home and interact with other people. The general public reported improved social acceptance and improvement in public attitudes toward people with mental health problem. The health professionals improved their receptiveness toward involvement of service users in developing and improving services as well as their attitude toward mental healthcare service delivery. |

PWLE = people with lived experience of mental health conditions.

members and caregivers while participating in the training, which led to increased mistrust and drop-outs in some cases (Rai et al., 2018).

Another challenge was the low literacy rate among PWLE, and perceived self-stigma induced low self-confidence among PWLE, making it difficult for them to participate in the intervention. Speaking in front of an audience, such as health workers, was anxiety-inducing, especially when asked intrusive questions. Further, recounting painful past experiences (lived experience and stigma experience) sometimes increased their distress.

### Strategies for overcoming challenges

1. Caregivers play an active role in all aspects of PWLE's treatment and recovery, including their daily activities. Without their involvement, it was difficult for PWLE to participate in the training and intervention. Hence, we actively involved caregivers in all aspects of participation, such as the consent process, participation in the training, and narrating recovery stories to health workers (Rai et al., 2018). This approach helped reduce drop-out from the program as they helped PWLE travel to and from training venues and manage time for the training.

2. Instead of viewing information sharing and consent as a one-time event, we stressed that this must be an ongoing process where we checked in on the PWLE and caregiver's understanding of the intervention in each session. Information sharing put PWLE and caregivers at ease during their early stages of involvement as they narrate recovery stories in front of an audience was a daunting concept for them without practice and preparation.

3. Training and preparation will be needed for PWLE before their involvement in anti-stigma interventions to help them understand the program and process better, which will equip them to deal with disclosure and distress.

4. As recalling and narrating past experiences of illness and stigma was distressing for some PWLEs, we involved psychosocial counselors throughout the training and intervention process. The counselors monitored any signs of distress among the PWLE and checked in on their psychosocial well-being. Their presence helped to reduce the concern of creating more distress and harm to PWLE.

5. The research team extensively discussed the process and consent before recruiting the PWLEs. In Nepal, we asked the PWLE to invite one caregiver (family member, close relative, or friend) who had disclosed their condition to the training. Hence, along with the PWLE, the caregiver was also involved throughout the training process. A counselor was present in each session of the photovoice training to check in with the PWLE and caregiver and to manage any distress that could come up during the training session. Each training session began with checking in and discussion with the participants regarding the benefits and harm of their participation in the program.

6. Disclosure of mental illness and the safety of PWLEs were key ethical concerns of our team. We involved a caregiver (a person PWLE felt comfortable with and consented to participate) in the entire training and intervention process. We also included safe disclosure and handling difficult question sessions in the photovoice training.

### Key learning points and recommendations

Caregivers can play a vital role in the involvement of PWLE in anti-stigma intervention programs – especially in a collectivist community like Nepal. Their involvement would ease PWLE to manage time, reducing external stress such as traveling to the training venues. It also helped in the involvement of PWLE with low levels of functioning.

Continuous information sharing and consenting processes where PWLEs are made aware that their involvement is voluntary, and it ensures safety and reduces intimidation among PWLEs. The involvement of mental health professionals throughout the process will help to ensure early detection and management of negative consequences of involvement or signs of distress.

### Case Study 2. A social contact-based education intervention to reduce stigma among community healthcare staff in Beijing, China

#### Background to the work

Peking University Sixth Hospital in Beijing, China, focuses on clinical treatments, practitioner training, and mental health research. In Chinese culture, there is significant stigma associated with mental health issues, affecting both the general public and community mental healthcare staff (Lam et al., 2010; Lv et al., 2013; Li et al., 2014). A collaborative initiative with King's College London, "Reducing Mental Health Stigma in Chinese Healthcare Practitioners" took place from 2016 to 2019. In this groundbreaking study involving 4 people with lived experience (PWLE) and 121 community mental health workers, PWLE were actively engaged for the first time in mainland China's mental health stigma reduction interventions (Zhang et al., 2022). There were no monetary benefits to the participants (PWLEs) during the training. However, during the social contact, each PWLE received 300 yuan labor fee.

The study was approved by the Peking University Institutional Review Board (Year 2017, No. IRB00001052-16077).

#### Anticipated challenges

These comprised identifying PWLE who fulfilled the inclusion criteria – having a full-time job, good at storytelling, able to speak fluently at a moderate pace, not exhibiting obvious side-effects of treatment and getting good narratives without further worsening stigma was expected to be difficult. Moreover, PWLE would be potentially apprehensive about disclosing their stories to the community workers and may not be intrinsically motivated in mental health stigma reduction activities.

#### Experienced challenges

When the research team identified suitable PWLEs, many refused to participate because training was normally held on weekdays and most had difficulty applying for medical leave as their condition was often concealed from their managers and colleagues.

The initial attempts of PWLE to write and narrate stories using the recovery story checklist were characterized by a tendency to record only the treatment process rather than the impact of their mental health conditions on their personal and family lives, or their feelings in the recovery process. During the intervention implementation, PWLE felt anxious speaking in front of community staff and expressed signs of worry.

#### Strategies to overcome challenges

To overcome challenges, the inclusion criteria were widened to include PWLE with different work statuses. PWLE known by the research team were approached first, followed by PWLE who had past experience of stigma and discrimination and had demonstrated a strong motivation to help reduce mental health stigma and discrimination.

PWLE and their family members were supported by psychiatrists, psychotherapists, general practitioners and social workers throughout the intervention. Measures to avoid pictures/videos were taken unless prior consent was obtained.

During the initial training, PWLEs wrote their stories with the help of family members and the research team. Further, each PWLE rehearsed their stories and received feedback from the research team. At implementation, PWLE were allowed to read from the script if they forgot any part of their story, and they were accompanied by family members who could provide additional perspectives on the stories.

Research team discussed disclosure and answered any questions with PWLEs and their family members/caregivers before disclosure decisions were made. PWLEs were monitored for potential harm over time through continued professional contact with the lead organizer. The communities in which PWLE participated in the training were not the localities in which they lived.

To support PWLEs' involvement, the local ethics board suggested involving family members.

#### Key learning points and recommendations

We found that PWLE cherished these opportunities and took the anti-stigma intervention seriously, arriving early at the research site

every time and feeling satisfied with themselves after completing their work. Their testimonies encouraged the team to continue such endeavors in the future. Interventions of this kind can empower PWLE to strengthen their confidence. The work also showed that it is possible to recruit participants who are willing to share their recovery stories, and contribute toward reducing mental health stigma.

### Case Study 3. Experience from "Stories Against Stigma" – a Walking Tour of a Tertiary Mental Health Care Facility, Bengaluru, India

#### Background to the work

The National Institute of Mental Health and Neurosciences (NIMHANS), established in the mid-19th century, provides modern training, research, and mental health services. Despite commendable efforts to raise awareness through group sessions, TV programs, and publications, stigma persists. PWLE face stigma affecting various aspects of their lives. To combat this situation, NIMHANS introduced a groundbreaking "Walking Tour" in 2018, inviting the public to interact with mental health professionals, explore the institute, and hear PWLE share their recovery stories. Two PWLEs participated, and the event garnered positive feedback from the public and was featured in national media (Meena et al., 2023).

One of the PWLE who took part was offered travel allowance, and another was admitted as an inpatient and was managing well during the tour, hence no compensation was provided.

#### Anticipated challenges

The team anticipated that involving PWLE is sensitive because they and their family members often fear the disclosure of their mental health condition to the community. They worry that disclosure might lead to losing their job, detrimentally affecting marriage prospects or exclusion from their extended family members. Apart from our experience, we were also uncertain of PWLE's perspectives about participating in a tour and the training process.

#### Experienced challenges

PWLE with active symptoms were unable to provide informed consent or reflect on/represent recovery. Therefore, an experienced psychiatrist was involved in identifying and inviting individuals to participate in the tour. Another difficulty was obtaining consent from family members of PWLE who were willing to take part in this public event. The families were uncomfortable with the idea of the person speaking openly about illness experience and recovery.

#### Strategies for overcoming challenges

To overcome the above-mentioned challenges, recovered and highly motivated individuals who had indicated their interest in raising awareness during their clinical visits were considered. A therapeutic and working relationship with the treating psychiatrist was crucial to garner the trust of PWLE. In addition, the long-standing cordial relationship, rapport and ensuring trust and confidence among the PWLE and their family members strengthened their participation.

The team discussed disclosure and answered any questions regarding this with PWLEs and their family members/caregivers before making disclosure decisions. As per the caregivers' request, disclosures were pseudonymized (not using the person's real name or photo). Opportunities to debrief about disclosure were offered to PWLEs and their family members/caregivers. PWLEs were monitored for potential harm over time through continued professional contact with the lead organizer.

Sharing recovery stories without pictures and names of PWLE in the public domain alleviated concerns of their family members. Two service users participated in the walking tour, and the treating psychiatrist carried out a pre-event briefing to explain the nature and expectations of the tour. The narratives of PWLE were edited for brevity and to emphasize the recovery process.

The walking tour was a first-of-a-kind initiative taken by the institute and having PWLE come in and share their narratives during the tour was suggested by psychiatrists who were also part of organizing the tour. The entire tour was carried out under the supervision of mental health professionals, including psychiatrists and psychiatric social workers, who suggested involving family members along with PWLEs.

#### Key learning points and recommendations

It was easy to work with motivated and trained volunteers to deliver their recovery stories, which was crucial. Since PWLEs do not necessarily have experience in delivering their recovery stories in public, they may express many details about their lived experience and can get emotionally distressed. Understandably, this situation requires empathetic listening and validation from the team, but at the same time, the content of their stories needs to be carefully edited to make a strong and meaningful impact on the audience and to minimize over-disclosure and distress.

We are considering PWLE to take the initiative to carry out future tours and provide one-to-one interaction with the public. However, the confidentiality of the individual, cultural norms and family concerns must be acknowledged and respected. Strategies such as keeping the media informed about what to report (using proxy names & not publishing their pictures), and debriefing sessions for PWLE after the event, are useful in addressing such concerns. The team suggests offering appropriate remuneration to PWLE and the caregivers for their time and travel along with providing them insights about the impact of their work through discussion during booster sessions afterwards, which would make them feel significant.

### Case study 4. Using participatory action research for service user and caregiver involvement in mental health system strengthening in primary care in Ethiopia

#### Background to the work

In the Ethiopian mental health system, involving PWLEs is challenging due to stigma and lack of awareness about experiences of and impact of mental distress. To address this situation, researchers from Addis Ababa University conducted participatory action research between April 2017 and August 2019 in Sodo district, south-central Ethiopia. Service users, including people with schizophrenia, epilepsy, alcohol use disorder, or depression, formed a Research Participant Group (RPG). They underwent 10 days of face-to-face empowerment training based on "Emerging mental health systems in low-and-middle-income countries" (Emerald) empowerment manual (Abayneh et al., 2017a) and the RESHAPE curriculum, receiving compensation of 300 Ethiopian Birr per session, along with tea and food during breaks (Abayneh et al., 2017b, 2020, 2022a, 2022b, Lempp et al., 2018).

Each participant was compensated with 300 Ethiopian Birr per session.

Ethical approval was obtained from the Institutional Review Board of Addis Ababa University College of Health Sciences (Protocol number: 027/16/Psy).

### Anticipated challenges

Although we intended to recruit recovered service users, there was a concern about relapse, lack of adequate transport and long-distance travel, particularly for service users coming from rural areas could affect participation. Stigma and low community expectations of what PWLE could meaningfully contribute were expected to be an impediment to PWLE's willingness and confidence to actively participate and openly share their lived experiences. Heterogeneity in the PWLE group, for example, diagnosis, gender, or educational level, might have undermined common understandings because of their different experiences of recovery.

### Experienced challenges

During the participatory action research (PAR) process, staff noticed that some PWLE became distressed while sharing their lived experiences of mental health stigma and discrimination, challenges with accessing mental healthcare, and associated costs and social burdens. The low educational level of service users meant that they could not fully engage in activities that involved writing. Many of the experienced challenges were related to sustaining PWLE involvement, and people coming from rural areas during the rainy season struggled to attend meeting sessions on time. The PWLE had concerns about operational expectations to sustainable involvement beyond the life of the project, including where they would meet and adequate funding to cover engagement-related costs. They also mentioned systemic and organizational constraints (e.g., lack of national-level regulatory mechanisms to enforce service-user involvement in mental health policy and mental health legislation; Abayneh et al., 2022a).

### Strategies for overcoming challenges

We identified the following key strategies for overcoming challenges, such as engaging with PWLE with varying levels of literacy. For example, with participants from rural areas with only basic levels of literacy, programmes utilized natural discussion rather than PowerPoint presentations or didactic lectures, as many PWLE were not used to engaging with information through these means. This helped the team to make the training and PAR process more inclusive and participatory. When writing was required, participants were assisted by family members and research assistants. Their caregivers also supplemented the discussion process by providing additional explanations about PWLE's stories.

Equipping PWLE for participation ahead of time (through photovoice and empowerment training; Abayneh et al., 2022b) was important. In addition, they mentioned that active and energetic facilitation of the PAR process helped to obtain their active engagement and involvement throughout the activities and sessions (Abayneh et al., 2022a).

Pre-planned strategies included the following: (i) preparatory training of participants on crisis management, identifying and capitalizing one's inner strengths, and (ii) the research process was facilitated by a mental health professional with extensive experience working with people with mental health problems. Arrangement of a convenient, central place and time (during the weekend or holidays) for the training and PAR processes, which assisted to maximize participation, was essential.

There was critical reflection with the research participants and the research team at the beginning and end of every day. In addition, two psychosocial professionals (one male and one female) attended throughout the study to provide any needed support, which was of value in our research to handle some actual challenges, for example, causal attribution between a caregiver and service user about illness.

The research team was conscious of the vulnerability of service users and arranged the necessary capacity-building and safeguarding measures before and during working with PWLEs.

### Key learning points and recommendations

Making use of participatory strategies in the research process was a key mechanism to supporting PWLE involvement, including establishing the small RPG that met regularly, utilized techniques to ensure everyone's voice was heard and the commitment to involvement from the beginning to set research priorities. The narrative of lived experiences was found to require careful planning. Preparatory training for strategic disclosure and crisis management for PWLE, as well as necessary precautions in the selection of the content of recovery stories to be adapted to ensure the participants did not feel distressed during the intervention process, was an important consideration.

The combined use of visual data (photos) together with individual and group reflective sessions (voice) can facilitate genuine, active participation, co-creating/co-production of knowledge, and provides a powerful means for marginalized groups to communicate their lived experiences.

The PWLE expressed satisfaction with their personal involvement, valued the experiences gained from the PAR process, and demanded various support for greater and sustained contribution. This initiation led to the establishment of a service user association and a strengthened voice of the PWLE in the community.

## Discussion

In this paper, we have presented experiences in the form of case studies from various anti-stigma programs implemented in four different countries (China, Nepal, India, and Ethiopia) of LMICs. In general, all these programs had active involvement of PWLE. These sessions were held at places where stigma toward mental illness was a major concern, and was influenced by culture (Loganathan and Murthy, 2011, 2008). For example, Chinese culture had a high level of stigma related to mental disorders (Phillips et al., 2002; Lv et al., 2013). Additionally, the involvement of a PWLE in stigma reduction is a new concept in most of the case studies, which is similar to existing evidence that despite interest in achieving greater involvement of PWLE, experiences and awareness of practical applications of this kind of engagement is limited in LMIC countries (Semrau et al., 2016).

The RESHAPE module was commonly applied to train PWLE in three case studies (China, Nepal, Ethiopia), and they received photovoice training, where individuals narrate their recoveries with a photograph (Kohrt et al., 2020, 2021). Most of the research programs summarized in these case studies found that direct contact with the public had positive effect in reducing stigma and discrimination, as outlined in Table 1. This is similar to existing evidence that the positive effect was stronger with direct contact (London and Evans-Lacko, 2010; Nguyen et al., 2012; Stubbs, 2014; Nyblade et al., 2019).

In all case studies, the program coordinating team anticipated that identifying PWLE would be difficult because their participation needs time, which may disrupt their household, jobs, or other

daily chores. Another anticipated challenge was that PWLE would be apprehensive about disclosing their stories of mental illness because of the negative consequences and experience of stigma, similar to what has been reported as potential barriers by previous studies (Lempp et al., 2018; Thornicroft et al., 2022). Cultural diversity and lifestyle differ among people from LMIC and high-income countries. Thus, during the process of planning and implementation, the team encountered minimal involvement of the PWLE due to a lack of empowerment and uncertainty about engaging PWLE (Petersen et al., 2017; Kohrt et al., 2021). PWLE needs to be involved from the beginning in leadership positions wherever possible (Thornicroft et al., 2022). Given the lack of experience of involvement in our countries, our efforts fell short of this recommendation. The authors intend that these case reports can help to catalyze the greater and meaningful involvement of PWLE. However, each country team recognizes that this will need explicit attention to reduce power differentials and create safe spaces for learning and development (Thornicroft et al., 2022).

In addition, the identification of suitable PWLE, refusal to take part due to work and fear of disclosure were the major challenges. However, service users had to make excuses to their families and caregivers while involved in the training, which led to increased mistrust and dropouts in some cases, as reported in the study by Rai et al. (2018).

The families were uncomfortable with the idea of the person speaking out openly about their mental illness experiences and recovery (India), similar to earlier study results, such as concerns about marriage, autonomy, social devaluation, fear of rejection, uneasiness about disclosure, feelings of shame and embarrassment about their condition. These were identified as factors having an adverse social impact of the illness on the person diagnosed and their caregivers (Thara and Srinivasan, 2000; Charles et al., 2007).

During the initial phase of the intervention implementation, PWLE felt anxious and expressed signs of worry as they were uneasy about their ability to handle the new situation, commonly linked with a low literacy rate, perceived self-stigma and low self-confidence. However, better support and time to develop trusting relationships in Ethiopia, for example, PWLEs gained their confidence (Abayneh et al., 2022b), which is in line with the existing finding that self-stigma can be reduced when the person with mental illness recovers and shares his/her experiences of illness and the successful recovery process (Corrigan et al., 2013). In LMICs, the components, attributing factors, focus, and implementation processes of various anti-stigma interventions varied based on their feasibility and acceptability in the respective social and cultural environments (Hanlon, 2017). Further, PWLE were asked to recall their lived experience with mental health conditions and stigma (Nepal and Ethiopia). This recounting of painful past experiences sometimes increased their distress, expressing the need for appropriate support and adequate preparation (Kaiser et al., 2020).

Different strategies were employed to overcome challenges involving PWLE in stigma reduction toward mental health care. Service users with varied educational backgrounds, who recovered, with past stigma experience and strong motivation to reduce stigma, were considered. PWLE known by the research team were approached first due to long-standing cordial relationships, rapport, trust, and confidence among the service users and their family members, which helped the participants (China, India) in similar with the guiding principles for the involvement of lived experience in decision-making (Sartor, 2023). Continued encouragement, information sharing, debriefing and support for

family members by the multidisciplinary team play an essential role in the involvement of PWLE in mental health conditions (China, India).

Furthermore, having a psychosocial counselor or mental health professionals throughout the training was useful when PWLE became distressed due to recalling past painful experiences (Nepal, Ethiopia; Abayneh et al., 2020). Similar to the existing work by Rai et al. (2018), writing recovery narrations, family members' help, rehearsal of stories and regular training was crucial (narration), and the involvement of family members (China, Nepal, Ethiopia) helped in reducing dropout from the program. Structuring, planning and delivery of contact situations influence the effectiveness of the intervention (Chen et al., 2016). The quality of professional contact included in the intervention plays a significant role in improving relationships with people with lived experience (Carrara et al., 2021). Speakers, messages, and interactions are found to be the primary constructs in designing the contact situation, similar to the existing literature by Stuart et al. (2014), Chen et al. (2016) and Stuart (2016). A convenient place and time (during the weekend or holidays) for the training was found to be necessary.

The case studies found that PWLE were willing to participate and valued the opportunities to get involved in anti-stigma activities. It also strengthened their confidence, and they felt satisfied after the training, which was similar to what was found by (Abayneh et al., 2022b). Caregivers of PWLE's also play a vital role in the anti-stigma intervention programs by managing time and reducing external stress. Empathetic listening and validation by the team, and the content of their stories need to be carefully edited to make a strong and meaningful impact among the audience and to minimize over-disclosure and distress (India and Ethiopia) was in line with structured testimonials that are important in stigma reduction (Kohrt et al., 2021). The author recommends that the confidentiality of the individual, cultural norms and family concerns be prioritized and respected during the implementation of the activity. In China, PWLE would not be sharing their stories in their own community or the district where they were living so that the trainees (community mental health workers) would not know or interact with them later in their lives.

Strategies like informing the media about what to report (e.g., using proxy names and not publishing PWLE's pictures) and debriefing sessions for PWLE are useful in addressing such concerns (Thornicroft et al., 2022). Offering appropriate remuneration to PWLE and the caregivers for their time and travel and providing them insights about the impact of their work would make them feel significant (Thornicroft et al., 2022; Sartor, 2023). Combined visual data (photos) and individual and group reflective sessions (voice) can facilitate genuine, active participation, co-creating of knowledge, and provide a powerful means for marginalized groups to communicate their lived experiences. In addition, the findings support the need for an explicit "transformative shift" of attitude toward global mental health, as mentioned by (Ryan et al., 2019). The study outcome provides further knowledge published by (Semrau et al., 2016), namely that the involvement of service users can take place at various levels, for example, in training healthcare personnel on mental health, peer review of policy planning and implementation.

These barriers, facilitators and strategies reported across the four study sites in all case studies are summarized in Table 2.

As the evidence base for collaborating with PWLE in stigma-reduction activities in LMICs is growing, a valuable next step would be for programs to routinely carry out implementation and process evaluations of this work. This would yield standardized insights on

**Table 2.** Barriers, facilitators and strategies in establishing partnership with PWLEs

| Barriers | Facilitators | Strategies to mitigate barriers/strengthen facilitators |
|---|---|---|
| • Refusal to Participate*<br>• Discontinued or dropout* | • Recovery stage*<br>• Volunteer**<br>• Motivation* | • Involved PWLE who are doing well, or symptoms are well controlled with their treatment and attained recovery stage****<br>• Identify PWLEs through known contacts who would be willing to be engaged***<br>• Identify the interested PWLEs through engaging with community health workers and local psychiatric facilities (Government and private)*<br>• As per the caregivers' request, disclosures were pseudonymized (not using the person's real name or photo)*<br>• The team discussed disclosure and answered any questions with PWLEs and their family members/caregivers before disclosure decisions were made***<br>• Thanking the PWLEs* |
| • PWLE employment status* | • Flexibility in work pattern* | • PWLEs known by the research team were approached**<br>• Planned training on weekends/ holidays* |
| • Self–stigma**<br>• Low self–esteem/ confidence* | • Strong motivation* | • Involvement of family members or caregivers in training***<br>• There was critical reflection on self with research participants and the research team at the beginning and end of every day***. |
| • Fear of disclosure of mental illness in their own community/ district***<br>• Fear of negative consequence of disclosure of mental illness** | • Opportunity to disclose* | • Planned disclosure and Pre–event briefing***<br>• Each session of the training began with checking in and discussion with the participants regarding the benefits and harm of their participation in the program*<br>• The communities that PWLE participate in training were not the communities in which they live*<br>• PWLEs were monitored for any potential harm over time through continued professional contact with the lead organizer**<br>• Cultural norms and family concerns need to be prioritized and respected* |
| • Confusion about what to share or what not to reveal about mental illness*<br>• Lack of public speaking skills (worry & anxiousness)*<br>• Lack of public speaking skills*<br>• Limited experience in public speaking* | • Readiness and commitment to participate in social contact*<br>• Existing training module** | • PWLE wrote their stories about mental illness with the help of family members*<br>• Family members helped in the narration of recovery stories***<br>• Adequate training to prepare recovery narration and disclosure of mental illness***<br>• Review of PWLE motivation and confidence and debrief from the research team*<br>• Use of photographs for recovery narration**<br>• Trained PWLEs on disclosure, recovery narration and public speaking skills, and team ensured PWLEs disclosed at varied levels and practiced before presenting to general public** |
| • Low level of education** | • Different level of literacy* | • Utilization of discussion and participatory approach in training PWLEs** |
| • Family stigma**<br>• Lack of support from family** | • Family support system*<br>• Trust and confidence with a MHP**<br>• Long–standing cordial relationship* | • Information sharing and consent to participate in social contact**<br>• Asked the PWLE to invite one caregiver (family member, close relative, or friend) who had disclosed their condition to the training session*<br>• Avoid pictures and names of PWLEs*<br>• PWLEs were monitored for any potential harm over time through continued professional contact with the lead organizer****<br>• Training team (Mental health professionals) psychoeducating the family members or caregivers on PWLEs' mental health problems to reduce family stigma*<br>• Encouraging caregivers to be involved in the training and participate along with the PWLEs in community programs**. |
| • Recounting painful experience**<br>• Emotional breakdown* | • Meaningful story about lived experience*<br>• Appropriate content for recovery story* | • Support from a mental health professional (Empathetic listening and validation)**<br>• Minimize over disclosure*<br>• A counselor was present in each session of the photovoice training to check in with the PWLE and caregiver and to manage any distress during the training session**<br>• Group therapy format was adopted for each session where the participants reflected on their issues and ensured peer support for each other and ensured therapeutic benefits* |
| • Travel and transportation** | • Session to be held in mutually convenient place** | • Honorarium for participation**<br>• PWLE is accompanied by caregiver***<br>• Organizing training in a convenient or central place and time*** |

*(Continued)*

**Table 2.** (*Continued*)

| Barriers | Facilitators | Strategies to mitigate barriers/strengthen facilitators |
|---|---|---|
| • Active symptoms/presenting symptoms of mental illness* <br> • Relapse of mental illness* | • Various diagnoses of mental health problems* | • Mental health experts are involved throughout the training and disclosure process, which helped research team to identify early sign/symptoms of mental illness and provide appropriate treatment and decide on when PWLEs can be involved in the stigma reduction activity**** <br> • Careful planning of training sessions for PWLE** <br> • Preparatory training on crisis management and identifying strengths is provided to PWLEs and their caregivers* |
| • Other | | • Meaningful engagement of PWLEs at the initial period when planning and developing a stigma reduction program*** <br> • Disseminating the lived experience recovery story through audio/video materials are another best strategy for stigma reduction in LMICs*** |

Element present in *one case study; ** two case studies; ***three case studies; ****four case studies.

practical barriers and facilitators of implementation, and the role of contextual factors, which could inform the development of step models to guide best practice in the field. An important consideration of such guidance would be how to assess when the potential harms of PWLE involvement could outweigh its benefits, and how to flexibly plan activities so that this risk is mitigated.

### Strengths and limitations

We present case studies from a diverse range of settings, reflecting different PWLE contexts. PWLEs were involved in these initiatives because the existence of stigma and discrimination toward PWLEs in their culture. Another strength is that PWLEs were trained using a standard and established module in LMICs (RESHAPE) across 3 case studies, which represents the quality of social contact. On the other hand, we present the findings from four case studies from different countries, with insightful reflections and these allow for comparisons between different contexts. Case studies are not generalizable; however, they illustrate important learning and detailed insight from different settings with similar and different perceptions. Any research or anti-stigma activities will have its strengths and limitations in the implementation process, which need to be taken into account when interpreting the findings. Considering this work as an important start in this area of research, we acknowledge and are aware of the potential critiques of our work, that is, the initiatives were not led by PWLE, clinicians/clinical settings were predominant in selecting the PWLE in sharing individuals' recovery narration.

The authors have implemented the social contact strategy in each setting to reduce power differentials and equip PWLE to take on and commit to increasingly active roles in the anti-stigma activities process. Additionally, a key limitation is the lack of involvement of PWLE in writing up this paper as the case studies were obtained from different countries and the PWLE were from varied educational backgrounds.

### Conclusions

Involving PWLE to reduce mental health-related stigma and discrimination is an essential strategy worldwide. Despite the sociocultural barriers, it is possible to involve PWLEs in social contact-based anti-stigma interventions and other mental health activities. Training PWLEs to narrate their condition or recovery and build skills to facilitate social contact-based interventions is likely to enhance the intervention's effectiveness. Developing or adapting cultural and contextual-based training modules to train PWLE is crucial. Debriefing and providing adequate information about the training, intervention, and role of PWLE and their family in anti-stigma intervention will highlight the value of their involvement and its impact at the community level or the general public. Involving multi-disciplinary mental health staff throughout the process of recruitment and training is important, as is PWLEs leading or co-leading anti-stigma initiatives, alongside support to reduce the risk of disclosure consequences and emotional distress or discomfort due to lived experience sharing.

**Open peer review.** To view the open peer review materials for this article, please visit http://doi.org/10.1017/gmh.2024.69.

**Data availability statement.** The case study reflections synthesized in the current study can be made available upon reasonable request to the corresponding author.

**Author contribution.** G.B.M. led the writing of the article. G.B.M., P.C.G., B.A.K. and C.H. conceptualized the article. P.C.G. conceptualized the methodology and provided modification. W.Z., S.A., D.G., C.H., S.L. provided site case studies and reviewed the full article. P.C.G., B.A.K., H.L., G.T. shaped the article and reviewed it multiple times, provided expert opinions and reading materials. All authors have reviewed and approved the final submission of this work.

**Financial support.** This research received no specific grant from any funding agency, commercial or not-for-profit sectors. W.Z. is supported by Beijing Health Technologies Promotion Program (BHTPP2022027). P.C.G. is supported by the UK Medical Research Council (UKRI) for the Indigo Partnership (MR/R023697/1) award. G.T. is supported by the National Institute for Health and Care Research (NIHR) Applied Research Collaboration South London (NIHR ARC South London) at King's College Hospital NHS Foundation Trust. The views expressed are those of the author(s) and not necessarily those of the NIHR or the Department of Health and Social Care. G.T. is also supported by the UK Medical Research Council (UKRI) for the Indigo Partnership (MR/R023697/1) awards. C.H. received support from the National Institute for Health and Care Research (NIHR) for the SPARK project (NIHR200842) and through the NIHR Global Health Research Group on Homelessness and Mental Health in Africa (HOPE; NIHR134325) using UK aid from the UK Government. C.H. also received support from WT grants 222,154/Z20/Z and 223,615/Z/21/Z. The views expressed in this publication are those of the authors and not necessarily those of the NIHR or the Department of Health and Social Care. For the purpose of open access, the author has applied a Creative Commons Attribution (CC BY) license (where permitted by UKRI, "Open Government License" or "Creative Commons Attribution No-derivatives (CC BY-ND)

license" may be stated instead) to any Author Accepted Author Manuscript version arising from this submission.

**Competing interest.** The authors have no conflicts of interest to declare.

**Ethics statement.** The process of collating reflections on the case studies reported on in this manuscript does not warrant ethical approval. Ethical approval was granted as relevant for the activities reported on in this manuscript at the respective institutions where the case study projects were carried out.

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
