## [Editor Report]

This is a very valuable and interesting manuscript to understand how initiatives and actions that are created in countries with greater resources that move to LMICs work, especially to incorporate PWLE in anti-stigma interventions. Therefore, a greater expansion in the description is requested. specially of some aspects, which would make it more complete and instructive for readers.

1. It would be important to know what types of stigmas that were targeted by the various programs (public, self, family, structural) and how this was done. A summary table (perhaps a supplemental table) would be helpful for the reader to understand the range of initiatives that could form a venue for contact-based education.

2. The paper indicates that involving people with mental illnesses in anti-stigma effort is important, both because it is a best practice in high income countries, but also because it can empower those who are involved. However, the paper also notes considerable difficulty in engaging people because of their fear of stigma and its significant consequences for their family and community life. How do the authors know that the involvement of PWLE in their efforts did not result in personal or family harms? Were participants followed? Any harms experienced would be important to elucidate prior to broadly recommending this approach. Also, I wonder if the potential for such harms would impact the ethics review process. Perhaps a comment on how this passed ethics review in each country and/or the lead site would be helpful.

3. Certainly, social contact has been demonstrated in high income countries to be an effective tool in disrupting public and self-stigma (not structural stigma). I would have liked to have greater rationale supporting the transferability of this approach to LMICs. Has there been any evidence from LMIC to support this approach? If not, why do the authors think this would be a useful avenue when public stigma and its consequences are so much more evident in LMICs?

4. The barriers identified in the manuscript and summarized in Table 1 are unique to LMICs, but many of the solutions seem to be more generic and used in anti-stigma efforts in higher income countries. Perhaps the authors could identify the approaches that are unique to work in LMIC’s and those that are transferable or transferred from broader anti-stigma work. In other words, the new insights captured from these case studies deserve to be highlighted in more detail.

5. Caregivers were identified as playing a key role. Can the authors comment on whether their involvement reduces family (or caregiver) related stigma. In high income countries stigma from friends and family are typically one of the most frequently identified areas of difficulty. We any of the approaches used successful in buffering this in LMIC’s?

6. On page 11, the authors state that most of the case studies found that direct contact with the public had positive effect. What made the difference between positive and negative or no effects and how was this judged?

7. It would have been helpful if the authors had developed a step model to give guidance for those interested in initiating anti-stigma initiatives to follow. As part of this model, I would hope to see some way to assess when contact should not be pursued (i.e., when the harms outweigh the benefits) and how this should be determined.

8. In the introduction section, the article refers to “direct and indirect social contact”; perhaps providing the readers with examples of what is direct social contact and what is indirect social contact. On page 11 there is an example of direct contact through by way of narrative recovery stories. Maybe elaborating/identifying on what indirect contact looks like?

9. A couple of writing details that could improve:

a) Under the heading “Anticipated challenges”- to add a full stop at the end of paragraph.

b) Under the heading “Key learning points and recommendations”- grammar to be amended from “ It was easy to work with motivated and trained volunteer to deliver their recovery stories, which was crucial” to “It was easy to work with ”a/ the“ motivated and trained volunteer”.

---

## [Editor Report]

The article is accepted in its current version. The reviewers consider it to be a good and clear manuscript, a valuable contribution to reducing mental health stigma in LMICs.